psychology

achievement, goals, priming, meta-analysis, publication bias

**Author for correspondence:**
David R. Shanks
e-mail: d.shanks@ucl.ac.uk

# Publication bias and low power in field studies on goal priming

David R. Shanks[1] and Miguel A. Vadillo[2]

[1]Division of Psychology and Language Sciences, University College London, 26 Bedford Way, London WC1H 0AP, UK
[2]Departamento de Psicología Básica, Universidad Autónoma de Madrid, Spain

DRS, 0000-0002-4600-6323; MAV, 0000-0001-8421-816X

Research on goal priming asks whether the subtle activation of an achievement goal can improve task performance. Studies in this domain employ a range of priming methods, such as surreptitiously displaying a photograph of an athlete winning a race, and a range of dependent variables including measures of creativity and workplace performance. Chen, Latham, Piccolo and Itzchakov (Chen *et al.* 2021 *J. Appl. Psychol.* **70**, 216–253) recently undertook a meta-analysis of this research and reported positive overall effects in both laboratory and field studies, with field studies yielding a moderate-to-large effect that was significantly larger than that obtained in laboratory experiments. We highlight a number of issues with Chen *et al.*'s selection of field studies and then report a new meta-analysis ($k = 13$, $N = 683$) that corrects these. The new meta-analysis reveals suggestive evidence of publication bias and low power in goal priming field studies. We conclude that the available evidence falls short of demonstrating goal priming effects in the workplace, and offer proposals for how future research can provide stronger tests.

## 1. Introduction

Few topics in behavioural research have generated as much controversy in recent years as the question of whether behaviour can be influenced by subtle 'primes' [1]. Behaviour priming research has a close connection to work on 'nudge' interventions [2] and typically involves measuring the effects on some performance measures of presenting a situational cue or signal that is superficially unrelated to the task. A general theoretical framework for behaviour priming views it as the automatic activation of mental representations in the absence of awareness, leading to influences on attitudes, judgements, goals and actions [3].

For example, in one of the earliest demonstrations, Srull & Wyer [4] first asked their participants to unscramble sets of words to form sentences that described hostile behaviours. When subsequently

asked to rate the personality of a man named Donald, whose behaviour as described in a brief vignette was ambiguously hostile, they rated him as more hostile than did participants who unscrambled neutral sentences. Another famous example (money priming) is the apparent modification of people's behaviour on a variety of measures following exposure to images of money, or to tasks that involve subtle activation of the concept of money. Vohs, Mead and Goode [5] reported that money priming causes people to work harder on difficult tasks and to become less willing to help others.

Apart from its theoretical implications (e.g. [3,6,7]), the importance of this research lies in the possibility of influencing behaviours in ways that align with individuals' conscious preferences. Using primes to increase healthy eating [8], pro-environmental behaviours [9], and so on has major social implications, and indeed this research has fed into public policy [10]. The problem, however, is that the robustness and reproducibility of many of these priming effects is under scrutiny. Not only have there been numerous 'typical' experiments failing to replicate influential priming effects, there have also been several pre-registered, high-powered multi-laboratory efforts with similar outcomes. For example, a major replication study [11] sought but failed to reproduce money priming in a very large-scale, multi-lab project and another failed to replicate Srull & Wyer's hostility priming effect [12].

In addition to these replication failures, meta-analyses have cast further doubt on many priming effects. For example, although Lodder et al. [13] found an overall small-to-medium-sized money priming effect (Hedges' $g = 0.31$, 95% CI [0.26, 0.36]) in a meta-analysis of 246 experiments, they also obtained clear evidence of publication bias, with effect sizes being larger in studies with smaller sample sizes (see also [14]). Furthermore, they found that 62% of all standard studies obtained positive results but only 11% of pre-registered ones did, and published studies yielded larger effects than unpublished ones. These patterns are consistent with studies employing small samples and finding non-significant results being harder to publish. Another meta-analysis [15,16] similarly found clear evidence of publication bias in 43 independent measures of another form of priming in which risk-taking, gambling and other potentially harmful behaviours are claimed to be increased by primes which activate evolutionary 'mating' motives (young male syndrome). In the demonstrable presence of reporting and publication biases [17], exploitation of 'researcher degrees of freedom' (RDF) [18–20], and inadequate power [21,22] in behavioural research, residual evidence for priming effects must be regarded as weak and requiring confirmation in large-scale, pre-registered studies that can exclude these and other sources of bias as an alternative explanation of the observed effects.

Against this background, Chen et al. [23] have performed an important service by conducting a meta-analysis of research on a form of priming, distinct from but related to those mentioned above, namely goal (or achievement) priming. Studies in this area examine the effects of subtle reminders of achievement or goals, such as a photograph of a woman winning a race, on various task performance measures. In one study, for example, showing this photograph to employees in a fund-raising call centre increased the amount of money they raised by nearly 30% [24]. Chen et al.'s meta-analysis of 23 studies ($n = 3,179$) reporting 40 effect sizes yielded an overall effect size (Cohen's $d$) of 0.45 [0.38, 0.53], implying that this form of priming is quite robust. They also found that the overall effect was moderated by the specificity of the goal, modality (visual versus linguistic prime) and research setting (laboratory versus field). Of particular interest is that field studies yielded significantly larger effects ($d = 0.68$ [0.55, 0.81], $k = 8$, total $N = 357$) than laboratory studies ($d = 0.42$ [0.34, 0.50]). Chen et al. ([23, p. 236]) concluded that their results 'clearly show that a primed goal, relative to a control condition, increases performance', and the meta-analysis seems to provide support for practical advice, such as the recommendation that 'when sending written materials to employees, a photograph should be included that denotes effective job performance' ([25, p. 410]).

# 2. Motivation for current research

On the other hand, there are a number of reasons to be cautious about these conclusions. Probably the most similar domain in which priming effects have been studied is the artificial surveillance (or 'watching eyes') literature. In a typical study, a photograph of a pair of watching eyes is placed at a location proximal to where individuals engage in some form of prosocial behaviour such as making a charitable donation or hand-washing. Despite many reports that artificial surveillance cues cause people to behave more prosocially, as if they are being watched by real people, a recent and comprehensive meta-analysis found no overall effect [26]. Although the behaviours studied in these two sub-fields differ (task performance versus prosociality), the priming induction is often quite similar (e.g. watching eyes versus a photograph of a woman winning a race). Because of this similarity, the apparent absence of priming in the artificial surveillance domain encourages careful

examination of the contrasting claim that goal priming is robust. Moreover, several attempts to replicate particular goal priming effects have been unsuccessful (e.g. [27,28]).

Another important motivation for the present work is that Chen *et al.* did not report any tests for publication bias among the studies they included, something explicitly recommended in the PRISMA guidelines for conducting and reporting meta-analyses [29]. This is particularly important because Chen *et al.*'s meta-analysis overlaps quite considerably with a much larger one undertaken by Weingarten *et al.* [30] which did find evidence of publication bias in goal priming research. The likelihood of bias is highlighted by the fact that Chen *et al.*'s estimate of the effect of implicit goal primes in field studies ($d = 0.68$) is not only larger than the average effect size in psychological research (approximately 0.40–0.50; (see [22,31])) but is also appreciably larger than the average effect of *explicit* goals. In a meta-analysis of 384 effect sizes ($N > 16,000$), Epton *et al.* [32] estimated the effect size of goal setting on behaviour change at exactly half this size ($d = 0.34$), suggesting at the very least that the former is an overestimate. Chen *et al.* [23] provided several reasons to justify their belief that this literature is free from reproducibility problems. However, if publication bias is present in the studies included in their meta-analysis, then the meta-analytic effect size needs to be adjusted, ideally by several bias-correction methods, in order to confirm this belief. In the absence of such a correction (if needed), it cannot be inferred that goal priming is a reliable and reproducible effect. Stated differently, bias-correction methods enable a sensitivity analysis on Chen *et al.*'s claimed findings.

Also, as elaborated below, Chen *et al.* did not include in their meta-analysis some studies that clearly meet their selection criteria and included others which do not. Again at variance with the PRISMA guidelines, they provided no information about a number of other important aspects of their meta-analysis. For instance they did not describe their precise search protocol[1], did not describe the statistical model they adopted for combining results (fixed-effect versus random effects), the software they used, or any measures of consistency/heterogeneity such as $I^2$. They did not report or make openly available a data table of the effect sizes underlying their meta-analytic estimates.[2] Meta-analysis is a complex technique [33] and without full and transparent reporting, other researchers may find it very difficult to reproduce the results of a meta-analysis [29,34]. Indeed Lakens *et al.* [35] found that 25% of meta-analyses randomly selected from prestigious psychology journals could not be reproduced at all, and many results in the remainder could not be reproduced. Failure to make the underlying data available also precludes any future updating of Chen *et al.*'s meta-analysis as new data become available: updating meta-analyses is an important tool for cumulative science [36].

In the light of these issues, we present a critical appraisal of their meta-analysis. We then report a revised meta-analysis, with the underlying data being openly available, of all field studies on goal priming that meet the appropriate inclusion/exclusion criteria. We explain below why the field studies, the main focus of this comment, are particularly important. To preview, this reveals clear evidence of publication bias and low power in goal priming field studies, leading us to conclude that the available evidence falls short of demonstrating goal priming effects in the workplace.

# 3. Inclusions and exclusions in Chen *et al.*'s overall meta-analysis

We argue that the principles underlying Chen *et al.*'s selection of studies for inclusion/exclusion are unclear, placing a question mark over the interpretation of their main meta-analysis (note that henceforth we refer to their main analysis, with $k = 40$ effects, as their 'overall' one). There are several strands to this argument. We begin with some simple cases that do not require careful scrutiny of their stated selection criteria.

Like Weingarten *et al.* [30] before them, one of the apparently successful studies included in Chen *et al.*'s meta-analysis was by Eitam *et al.* [37]. A high-powered, pre-registered and near-exact replication attempt of this particular study was conducted as part of the Reproducibility Project: Psychology [38] but was unsuccessful [39]. Chen *et al.* did not include this replication study.

---

[1]Chen *et al.* (p. 230) stated that they "searched databases such as PsycINFO… for empirical experiments using the following key words: prime, priming, primed, subconscious, nonconscious, unconscious, and performance. The search was conducted for experiments conducted between 2006 and 2019… In our initial screening, we examined all titles and abstracts for relevant studies." PsycINFO alone yields nearly a quarter of a million hits for the disjunction of these keywords. The rules by which the 'screening' process reduced so many reports down to an initial set of 52 are not described.

[2]We thank Chen and colleagues for sharing their dataset with us.

Chen *et al.* included several laboratory experiments by Itzchakov and Latham [25,40] in their meta-analysis. However, each of these articles also includes a field study (Experiment 4 in [25], Experiment 3 in [40]) which was omitted from the meta-analysis without a compelling justification.

In addition to these problematic decisions, Chen *et al.*'s meta-analysis included and excluded other studies in what appears to be an idiosyncratic fashion. For example, despite including nine effects on creativity, they omitted experiments by Zabelina *et al.* [41], Maltarich [42] and Marquardt [43]. In all of these experiments, achievement goals were primed using the same types of manipulations as in the remaining studies and the outcome was a measure of creativity. Even a more careful examination of the stated dependent variable inclusion criteria does not shed light on these idiosyncrasies:

> '*Organizationally relevant dependent variables.* We excluded studies that focused on dependent variables arguably irrelevant to organizational behavior such as neurophysiological or physiological measures…, and self predictions or intentions … We included articles that focused on: (a) job/task performance, (b) creativity, (c) motives…, and (d) (un)ethical and fairness behavior… we merged the experiments that focused on performance and creativity to test the average overall effect of priming achievement on performance (*k* = 34)' [23, pp. 230–231].

Although the heading suggests that the meta-analysis is restricted to 'organizationally relevant' dependent variables, this category is not further defined, and more importantly the subsequent list and its implementation are inconsistent with this restriction. For instance, many of the included studies employed highly constrained measures of creativity, such as listing uses for a common object such as a coat hanger. Creativity is certainly important in the workplace, but so are attention, planning, decision making, multi-tasking and a large set of other basic mental abilities, so it cannot be argued that creativity has any privileged relevance to organizational behaviour. Moreover, Chen *et al.* included in their meta-analysis a study with non-creativity, laboratory dependent measures. In the study already mentioned, Eitam *et al.* [37] primed their participants with achievement-related words and then required them to perform one of two implicit learning tasks. To highlight how far removed these tasks are from job performance, consider the serial reaction time task used in Eitam *et al.*'s second experiment. In this task, a target stimulus appears at one of four locations on a computer screen across 350 trials and the task was to press a button as fast as possible to indicate each location; the target followed a repeating sequence and the task permits learning of this sequence to be measured. A large literature exists on this form of perceptual-motor learning [44].

It is hard to see how this kind of research can fall within Chen *et al.*'s stated inclusion criteria, unless those criteria are essentially unconstrained (which is perhaps what *task performance* means in the quoted passage above). Almost everything psychologists ask participants in their research to do can be described as a 'task'. But then if one laboratory study measuring a basic perceptual-motor learning ability [37] meets the inclusion criteria, why don't similar studies? For instance, Hassin *et al.* [45] reported experiments in which the effects of goal priming on performance in the Wisconsin Card Sorting and Iowa Gambling Tasks were studied, and Capa *et al.* [46] examined effects of goal priming on another learning task involving educational materials. Chen *et al.* omitted both of these studies from their meta-analysis.

When the effects included in a meta-analysis are incomplete and lack coherence, interpretation of the derived meta-analytic effect size estimate is undermined. We contend that this is the case with Chen *et al.*'s overall meta-analysis, in the light of these question marks over their inclusion/exclusion decisions. These problems would potentially have been avoided if Chen *et al.* had followed key PRISMA guidelines [29] relating to precise specification of the eligibility criteria.

A response to these concerns might be to propose a modest revision of Chen *et al.*'s dataset, adding in the studies discussed above. But the studies we have cited are merely examples and not the result of a planned literature search. In reality, fixing these issues would require an entire new meta-analysis.

## 4. Chen *et al.*'s meta-analysis of field studies

Field studies in which priming effects are evaluated in applied settings obviously have particular importance. Although laboratory studies typically enable closer experimental control of extraneous variables, their generalizability beyond the laboratory is often unknown. Another reason for focusing on field experiments is that laboratory research on goal priming has already been extensively considered in Weingarten *et al.*'s [30] much larger meta-analysis. The major difference is that Weingarten *et al.* included only experiments that employed verbal primes whereas Chen *et al.* included both verbal and non-verbal (e.g. photograph) primes.

Chen *et al*. do not provide a definition of a 'field study' other than to frequently juxtapose this category with laboratory experiments. They do, however, say (p. 223) that 'Field experiments, with random assignment to conditions, are arguably the 'gold standard' in organizational psychology. This is because they yield findings with both internal and external validity that can be readily adopted by managers (Eden, 2017)'. In Eden's [47] review, field experiments are characterized as ones conducted 'among members of an organization fulfilling their organizational roles' (p. 99). A much more comprehensive definition and taxonomy of field experiments was developed by Harrison & List [48], who distinguished between laboratory experiments at one extreme and *natural field experiments* at the other. True field experiments take place in the context in which the participants naturally undertake the tasks of interest, and they are unaware that they are in an experiment. Between these extremes, *artefactual field experiments* are just laboratory experiments carried out with atypical participant pools, while *framed field experiments* move beyond artefactual ones in studying tasks that are natural to that field setting.

As we elaborate below, Chen *et al*.'s meta-analysis of field experiments is undermined by the same idiosyncratic selection problems described above. It excluded six studies which meet their own definition of the key features of a field study while at the same time including another effect which does not. In the context of a very small number of effects ($k = 8$), these selection decisions substantially distort the set of included effects.

We have already noted that—despite including laboratory experiments by Itzchakov & Latham [25,40]—Chen *et al*. omitted from their meta-analysis the two field experiments included in these reports (Experiment 4 in [25], Experiment 3 in [40]). Both of these were conducted in a call centre (like the majority of the included studies) and one of them employed exactly the same prime stimulus as in many of the included studies (see below). Both had a measure of job performance (customer satisfaction) as their dependent measures. Indeed the authors of these reports explicitly described them as field studies of goal priming effects. They comprehensively meet Chen *et al*.'s stated search criteria (including the time period, as one was published online in 2018 and the other in 2019).

This is not the only problematic omission. Recall that Shantz & Latham [24] reported that the amount of money raised by employees in a fund-raising call centre was increased by the presence of a photograph of a woman winning a race. It is clear that this is a natural field experiment: it took place in an organizational setting and the dependent measure was directly related to job performance. Chen *et al*. omitted from their meta-analysis, in contrast, a near-identical experiment conducted by Stajkovic *et al*. [49]. These omissions are surprising given the overlap in authorship. They also omitted one of three field experiments reported by Bipp *et al*. [50], despite including the other two and despite including all three of these effects in their overall meta-analysis. Lastly, two experiments on election canvassing by Lenoir & Matthews [51] were omitted.

The above definitions of field research are important because they demarcate the contexts and conditions in which a study can meaningfully be described as a field study. Against this backdrop, it is clearly inappropriate to include—as Chen *et al*. did—a study in which 'adults… were approached… one at a time, in a train/subway station in a large metropolitan city, on their way to work' and asked to think of uses for a coat hanger ([24, p. 11]). The participants were not members of an organization [47] nor was the measured behaviour part of a field setting—this was an artefactual field experiment in Harrison & List's [48] taxonomy. Thinking of uses for a coat hanger may be a valid measure of creativity but it is not a measure of workplace performance. Whether or not it was appropriate of Chen *et al*. to include this study, our point is that this should not be a matter of subjective interpretation. As the PRISMA guidelines [29] emphasize, the inclusion criteria should be sufficiently clear that anyone applying those criteria would reach the same decisions.

In summary, these issues concerning inclusion/exclusion mean that interpretation of Chen *et al*.'s meta-analysis of field studies is clouded, just like their overall meta-analysis. Importantly, we emphasize that the issue is not about the particular studies reviewed above and their inclusion/exclusion, it is about the explicit principles underlying Chen *et al*.'s selection of studies for inclusion/exclusion. We argue that Chen *et al*. have failed to present a meaningful synthesis and meta-analysis of the available body of field experiments on goal priming, and that the meta-analytic effect size they reported (Cohen's $d = 0.68$ [0.55, 0.81]) is therefore uninterpretable.

# 5. An updated meta-analysis of field studies on goal priming

Given the relatively small number of relevant studies, it is far more tractable to compile an updated set of effect sizes for field studies than it is to revise Chen *et al*.'s overall meta-analysis. Furthermore, our

primary interest is in the field experiments, for several reasons. We have already noted that Weingarten *et al.* [30] have conducted a much larger meta-analysis of laboratory goal priming research ($k = 143$). And as Chen *et al.* themselves note, the implications of field experiments are highly important, comprising as they do 'the 'gold standard' in organizational psychology' yielding 'findings with both internal and external validity' [23, p. 223]. Accordingly, the conclusion that 'field experiments suggest that priming is a cost-effective managerial technique for enhancing human resource effectiveness' [23, p. 229] has significant implications for the workplace.

We therefore updated Chen *et al.*'s meta-analysis of field studies. We included the omitted studies noted above and dropped the study inappropriately included [24]. Our selection method was hence to take Chen *et al.*'s sample of studies as our primary source and to revise it on the basis of internal scrutiny (all but one of the experiments we added came from sources cited by Chen *et al.* which provided other effects to their meta-analysis). We supplemented this inclusion protocol in two ways: we searched Web of Science (6/1/2020) using the term '(prime or priming or primed) and (subconscious or nonconscious or unconscious) and performance', an implementation of Chen *et al.*'s [23] search descriptor, for articles published from 2006 onwards; this search yielded the Lenoir & Matthews [51] article; and we conducted a cited-reference search of [24], the first study to report a goal priming effect in the field.

The final set of studies, described in table 1 ($k = 13$, total $N = 683$), comprises 63% more effects from nearly twice as many participants as Chen *et al.*'s meta-analysis. Note that although the number of effects is modest and limits the ability to explore moderators, it is actually more than the median number of studies ($k = 12$) included in meta-analyses in psychology [54]. One study employed words as the prime [49], one used a photograph of employees wearing headsets and answering calls [25], one ([52], context-specific prime) showed a photograph of employees telephoning potential donors and one used a photograph of an election canvasser knocking on a potential voter's door ([51], canvasser prime). The remainder used the same photograph of a female athlete (Sonia O'Sullivan) winning a race (the photograph is reproduced in [40], Fig. 1; [51], Fig. 2; [52], Fig. 1; and [24], Fig. 1).

All studies employed a between-subjects manipulation of prime type. Apart from one study in which the dependent measure was academic performance [50] and another which measured the number of addresses visited by election canvassers, all took place in call centre offices and measured a direct indicator of call centre job performance (e.g. amount of money raised from donors) as their dependent variables.

We determined effect sizes for all studies based on the relevant descriptive statistics (for details, see the complete dataset at https://osf.io/5cjzp). In all cases where required data were missing from the primary reports, the original authors kindly provided additional information. We were able to closely reproduce Chen *et al.*'s effect size calculations (mean absolute discrepancy in estimated Cohen's $d = 0.024$).

We conducted a random effects meta-analysis using the 'rma()' function in the R 'metafor' package [55]. The individual effect sizes are shown in table 1 and figure 1 shows the forest plot. Across all experiments, the mean effect size is $d = 0.64$, 95% CI [0.41, 0.88] and is slightly smaller than the effect reported by Chen *et al.* ($d = 0.68$)[3]. The effects are moderately heterogeneous, $\tau^2 = 0.11$, $I^2 = 61.4\%$, $Q(12) = 30.33$, $p = 0.0025$. From the bottom two rows of figure 2, it can been seen that even though our meta-analysis includes considerably more effects than Chen *et al.*'s, their confidence interval is narrower. The explanation of this paradox is that the small set of effects Chen *et al.* included yields an underestimate of the true heterogeneity. When we run a random effects meta-analysis on their dataset, we obtain $I^2 = 18.2\%$.

Although the updated meta-analysis yields an overall large mean effect, considerable caution should be exercised in interpreting it: the funnel plot in figure 2 reveals a striking small-study effect. The figure shows all effect sizes plotted against their standard errors (experiments with larger samples and hence smaller standard errors appear higher on the vertical axis), and plainly there is an inverse relationship between precision (standard error) and effect size. This relationship is confirmed by Egger's test for funnel plot asymmetry, $z = 4.31$, $p < 0.0001$.

---

[3]The dataset includes three non-independent pairs of effects in which a single control group is compared to two treatment groups [50–52] (note that the sample sizes in these control groups have only been counted once in the calculation of the total $N$ quoted above). To assess the impact of this non-independence, we ran a multi-level meta-analysis, nesting these dependent effects within experiments, via metafor's 'rma.mv()' function. This yielded a mean effect size of $d = 0.69$ [0.43, 0.94], close to the value from the standard univariate meta-analysis. For the two methods that can be adapted to multi-level data, the bias-corrected estimates reported below in table 2 were identical (PET) or very similar (PEESE: multilevel intercept = 0.01 [−0.28, 0.31]). We therefore focus on the univariate analysis in the remainder of the article.

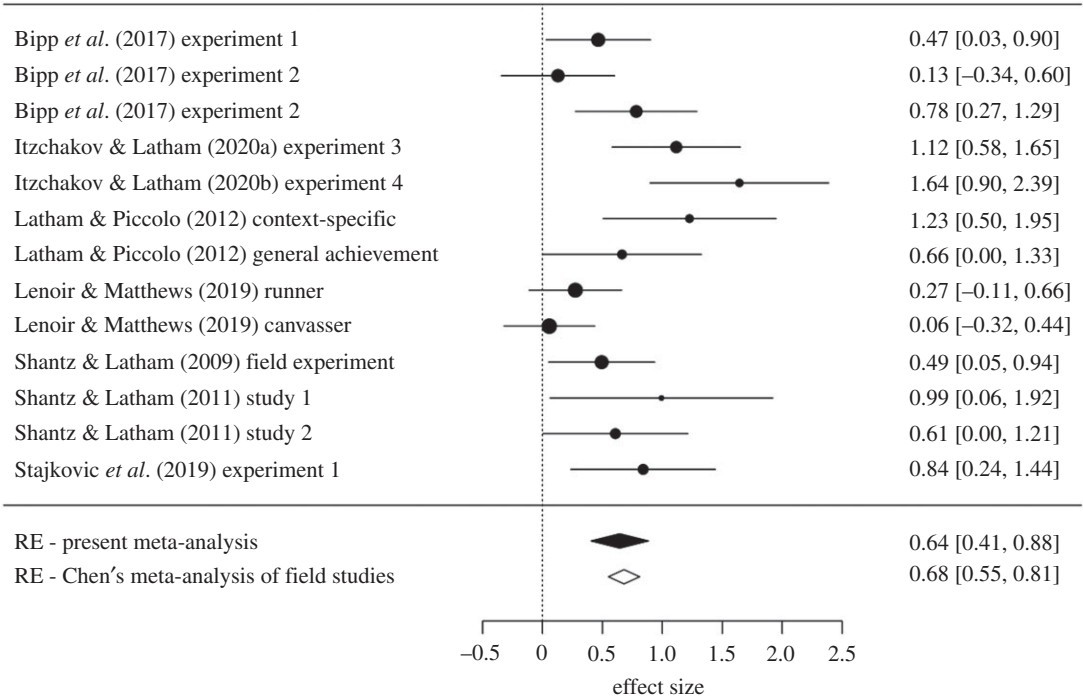

**Figure 1.** Forest plot. The rows denote the effect sizes (Cohen's *d*) and lower and upper 95% confidence intervals of goal priming effects. The final rows report the meta-analytic effect in the updated meta-analysis and in Chen *et al.*'s meta-analysis.

**Table 1.** Details of experiments included in the meta-analysis.

| study | prime type | dependent variable | N (prime) | N (control) | effect size (*d*) |
|---|---|---|---|---|---|
| Bipp *et al.* ([50], Exp. 1) | photograph | exam grade | 42 | 41 | 0.46 |
| Bipp *et al.* ([50], Exp. 2, photo runner) | photograph | exam grade | 33 | 36 | 0.13 |
| Bipp *et al.* ([50], Exp. 2, photo grade) | photograph | exam grade | 29 | 36 | 0.77 |
| Itzchakov & Latham ([25,40], Exp. 4) | photograph | customer satisfaction | 19 | 18 | 1.61 |
| Itzchakov & Latham ([25,40], Exp. 3) | photograph | customer satisfaction | 31 | 31 | 1.10 |
| Latham & Piccolo ([52], context-specific prime) | photograph | number of donor pledges | 17 | 18 | 1.20 |
| Latham & Piccolo ([52], general achievement prime) | photograph | number of donor pledges | 19 | 18 | 0.65 |
| Lenoir & Matthews ([51], runner) | photograph | canvassing | 51 | 53 | 0.27 |
| Lenoir & Matthews ([51], canvasser) | photograph | canvassing | 54 | 53 | 0.06 |
| Shantz & Latham ([24], Field Exp.) | photograph | money raised | 40.5 | 40.5 | 0.49 |
| Shantz & Latham ([53], Study 1) | photograph | money raised | 10 | 10 | 0.95 |
| Shantz & Latham ([53], Study 2) | photograph | money raised | 22 | 22 | 0.60 |
| Stajkovic *et al.* ([49], Exp. 1) | words | call-handling time | 23 | 23 | 0.82 |

Bias detection methods such as this are known to be generally underpowered [56], so this asymmetry is strongly suggestive of publication bias and/or the exploitation of RDF [20] in this literature. It is true that factors other than publication bias/RDFs (such as effect size heterogeneity) can cause small-study effects [57]. If researchers conduct pilot work to estimate the effect size likely to emerge in their main experiment, for example, and then allocate more participants to experiments exploring small effects, effect sizes and standard errors will be correlated even in the absence of selection bias. However, the studies included in the meta-analysis provide no reason to believe that such pilot work was undertaken. Most of the sample sizes are described as being determined by access constraints.

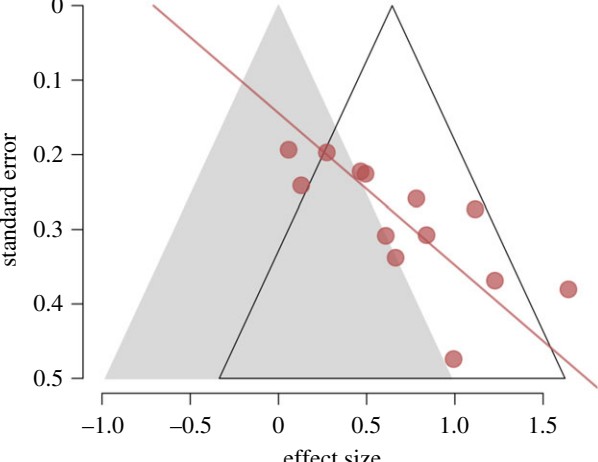

**Figure 2.** Funnel plot. Symbols represent the effect sizes (Cohen's *d*) of each experiment plotted against the inverse of that study's s.e. The line is the regression line from the Egger test, and the intercept is the PET estimate of the bias-corrected effect size. The shaded grey area depicts the region in which $p > 0.05$ for individual studies.

An example of an RDF is data exclusion. For instance, Bipp *et al.* ([50], Experiment 2) removed four participants from their analysis on the basis that their exam grades made them outliers, and these participants also reported low motivation. These exclusions may have been perfectly reasonable in the context of the research hypothesis, but in the absence of a full pre-registration of the planned experimental and analytic methods or an explicit disclosure statement (e.g. [58]) we cannot know whether these exclusion rules were formulated in advance or only after examining the data. Similarly, in their field experiment, Shantz & Latham [24] applied a square-root transformation to their dependent variable, a transformation applied in none of the other field experiments.[4] Again, this might be an entirely valid statistical procedure, but it might equally have been decided after examining the data. To be clear, we intend no aspersions on the integrity of these researchers— exploitation of RDFs is widespread across psychological researchers (ourselves included). The point is that these practices can shift an experimental result rightward in the space depicted in figure 2, possibly even taking it across the $p = 0.05$ boundary, hence contributing to the asymmetry of the funnel plot. This process can be additional to the selective reporting of statistically significant results, whereby studies that lie in the lower left region of figure 2 are missing from the scientific record.

A potential alternative interpretation of funnel plot asymmetry assumes that there may be systematic methodological or participant characteristics that generate the correlation between effects sizes and sample sizes.[5] It seems natural to enquire what features distinguish small studies with large effects from large studies with small effects. To test whether there are any systematic features that provide an alternative explanation of the funnel plot asymmetry, we coded all the experiments in the updated meta-analysis according to the theoretically motivated moderators included in Chen *et al.*'s [23] meta-analysis: whether the prime was specific or general, was visual or linguistic, and the time lag (seconds, minutes, hours, days) between presentation of the prime and the measured outcome. Details of these moderators can be found in Chen *et al.* [23], as well as the justifications of their hypotheses that priming would be greatest for specific, visual primes at longer lags. Our analyses show that none of these moderators made a significant difference to effect sizes, largest $Q(1) = 1.13$, $p$'s > 0.28. Furthermore, including these moderators in Egger's test for funnel plot asymmetry did not change the results: asymmetry remained significant in all cases, smallest $z = 3.89$, $p$'s < 0.0001. In other words, these moderators did not account for the funnel plot asymmetry. Particularly striking is that the majority (7/13) of studies employed a general/visual prime with a lag of hours, and even this small set of effects from studies with highly similar methods still showed distinct funnel plot asymmetry, $z = 1.73$, $p = 0.08$.

We submit that this analysis lends support to the publication bias conclusion. It might seem reasonable to ask whether there are features that distinguish small studies with large effects from

---

[4]Latham & Piccolo [52] and Shantz & Latham [59] used a logarithmic transformation.

[5]Details of the analysis described in this paragraph can be found at https://osf.io/5cjzp/.

large studies with small effects, but there is no set of such factors. The most straightforward explanation for the association is that studies with small samples and small effects have been excluded from the published literature. In other words, there is no evidence to suggest that the relationship in figure 2 says something about the underlying studies—it does not. The likely 'missing' studies—if they could be found—would almost certainly change the pattern of effects in the figure and the association would disappear.

# 6. Applying bias-correction methods to the updated meta-analysis

The regression line plotted in figure 2 suggests that an ideal study with a very large sample size (and standard error therefore close to zero) would obtain a negligible priming effect. But this is just one of several methods for correcting meta-analytic effect sizes for bias. There is a very long and rich literature on bias-correction methods (see [59], for a recent review; for an authoritative early source, see [60]). Among the many lines of evidence that these methods improve the accuracy of meta-analytic effect size estimates, they have been found to yield estimates that are closer to 'gold standard' effect size estimates obtained in pre-registered multiple-laboratory replication projects. Kvarven *et al.* [61] recently collated all available meta-analyses for which pre-registered experiments measuring the same effect have been conducted. These experiments precisely estimate effect sizes in the absence of publication bias. Every meta-analysis overestimated the effect size of its comparison pre-registered experiment, but this overestimation was reduced when bias-correction methods were applied to the meta-analytic effect estimates.

Unfortunately, it is clear that no single method is better than others in general, and moreover, the methods often do not concur on the presence or absence of publication bias [56,62,63]. This means that the typical practice of simply reporting the outcome of one or perhaps two such methods is unjustified. If the methods yield divergent outcomes, there is a significant risk of meta-analysts choosing to report only those correction methods that yield outcomes consistent with their theoretical predispositions. In this section, we describe a *sensitivity analysis* [62] that applies available methods in a principled way to minimize any possibility of this type of *p*-hacking of method application and reporting.

Validating bias-correction methods is not an easy task, as the true population effect size for the studies in a meta-analysis is by definition unknown. For this reason, simulation studies are essential because for these, both the true effect size and the underlying biases are known. In a comprehensive analysis, Carter *et al.* [62] simulated datasets typical of psychological research, varying the number of included studies, the true underlying effect size, the heterogeneity of the observed effects and the extent of simulated publication bias and RDFs. For each of over 400 combinations of these factors, data were simulated and several bias-correction methods fitted, and the proximity of the bias-corrected effect size estimate to the true effect size was recorded. Publication bias was modelled by calculating the *p*-value for a simulated study and retaining or rejecting it from the dataset on the basis of a range of more or less complex rules based on that *p*-value. RDFs were modelled by a range of rules that permitted different levels of outlier removal, optional stopping of data collection, switching between two dependent variables and so on.

These analyses yielded a complex pattern in which each of the methods worked satisfactorily in some conditions and unsatisfactorily in others. For example, focusing on datasets comprising $k = 10$ studies (the closest to the number in the present meta-analysis), the popular *p*-curve [64] and trim-and-fill [65] methods yielded false-positive rates (probability of falsely rejecting $H_0$ = no true effect) of over 50% when publication bias was strong and heterogeneity high, rendering them invalid methods in such circumstances.

Carter *et al.*'s analyses provide the motivation for a sensitivity approach in which the starting point is a description of the 'plausible' conditions that pertain in a meta-analysis, and using these conditions to constrain the choice of and weight given to bias-correction methods. For our meta-analysis, we define these plausible conditions as follows. First, the heterogeneity of the study effects is based on values from the two largest and most relevant meta-analyses of priming effects [13,30] which obtained values of $\tau = 0.34$ and $\tau = 0.38$, respectively, close to one of the simulation conditions ($\tau = 0.4$). Next, we assume that publication bias may lie anywhere between none and medium levels. It seems unlikely that publication bias is extreme in this literature, bearing in mind that the dataset includes effects that are not statistically significant. Lastly, we assume that RDFs span the full possible range between none and high levels.

**Table 2.** Meta-analytic estimates. Note: PET = precision-effect test; 3PSM = 3-parameter selection model; PEESE = precision-effect estimate with standard errors.

| method | mean effect size (*d*) | 95% CI | worst-case false positive rate (%) |
|---|---|---|---|
| random effects | 0.64 | 0.41, 0.88 | 63 |
| *methods identified as satisfactory by sensitivity analysis* | | | |
| PET | −0.71 | −1.30, −0.12 | 16 |
| 3PSM | 0.30 | −0.11, 0.70 | 14 |
| *methods identified as unsatisfactory by sensitivity analysis* | | | |
| trim-and-fill[a] | 0.40 | 0.12, 0.68 | 39 |
| PEESE | 0.03 | −0.29, 0.35 | 29 |
| PET-PEESE | 0.03 | −0.29, 0.35 | 22 |
| *p*-curve[b] | 0.64 | | 88 |
| *p*-uniform | 0.38 | −0.86, 0.89 | 63 |

[a]Trim-and-fill imputes five missing effects on the left of the funnel plot.
[b]*p*-curve does not provide a confidence interval on its effect size estimate.

Under these plausible and wide conditions, Carter *et al.*'s simulations (as derived from the online simulator available at http://www.shinyapps.org/apps/metaExplorer/; these are also provided in their Fig. 3) specify that only two methods, the precision-effect test (PET) and 3PSM selection model, perform satisfactorily in this context, which we define as returning a false-positive rate lower than 20%. The 20% threshold is a trade-off between conservatism and applicability. As we lower the threshold, we are less likely to make a false-positive inference, but it becomes more likely that all methods are rejected. All other methods yield higher false-positive rates under at least some combinations of the plausible conditions specified above. Hence we base our bias-corrected estimates on the PET and 3PSM models, although in table 2, for full transparency, we also report the corrected estimates of the other methods.

The PET test formally takes the intercept of the Egger regression (figure 2) as its estimate and when applied to our dataset yields a value of −0.71 [−1.30, −0.12]. We do not assign any significance to the fact that this is negative, which is presumably due to sampling error. We do attach significance, however, to the fact that the confidence interval is not above zero. On the basis of their simulations, Carter *et al.* [62, p. 134] concluded that 'A statistically significant PET-PEESE estimate in the unexpected direction probably is incorrect, but researchers should be aware that when they obtain such an estimate, there is likely to be some combination of QRPs and publication bias and, perhaps, a null effect'.

Selection models including 3PSM start from the assumption that studies may exist that have not been included in the meta-analysis and that it is a study's *p*-value and direction that determines the likelihood of inclusion. As a consequence, the mean observed effect size becomes artificially inflated. The Vevea & Hedges [66] 3PSM selection model corrects for the resulting inflation of effect sizes created by this selection process. This model assumes that the distribution of observed effect sizes depends not only on their mean effect and the heterogeneity across studies, but also on the probability that studies with non-significant results end up being published. This probability is modelled as an additional free parameter. When this model is applied to the updated set of field study effects, via the R 'weightr' package [67], it returns an adjusted effect size that is appreciably closer to zero than the random effects estimate and no longer statistically significant, $d = 0.30$, 95% CI [−0.11, 0.70], though the small number of studies means the precision of the estimate is very low. Importantly, the likelihood ratio comparing the fit of the selection model to that of a standard random effects meta-analysis is significant, $\chi^2(1) = 5.44$, $p = 0.020$. Hence, a model that allows for publication bias fits the data better than an unadjusted model.

Table 2 reports results from five other methods, trim-and-fill, PEESE, PET-PEESE, *p*-curve and *p*-uniform. Details of these methods can be found in [62]. Only two of these methods reject the null hypothesis: trim-and-fill yields an estimate with a confidence interval that excludes zero (though still appreciably lower than the random effects estimate, and imputing five missing effects), and *p*-curve (which in Carter *et al.*'s implementation does not generate a CI) yields a value identical to the random effects estimate. The rightmost column of table 2 details the basis on which these methods are

identified by the sensitivity analysis as producing unacceptably high (greater than 20%) false-positive rates under at least some plausible conditions. For example, the rate is nearly 90% for *p*-curve, meaning that for this percentage of randomly generated datasets, *p*-curve rejects the null hypothesis even when the true effect is zero.[6] Given that a false positive of 60% or so is likely, the results also emphasize that it is indefensible to draw conclusions from a standard, uncorrected random effects analysis, as Chen *et al.* did.[7]

In summary, a sensitivity analysis suggests that two bias-correction methods are acceptable in the plausible conditions of the meta-analysis, namely PET and 3PSM. When applied to the data, these methods yield much lower estimates of the overall effect size and indeed indicate that the null hypothesis that there is no true priming effect cannot be rejected (since zero is included in the 95% confidence intervals of both estimates). The overwhelming likelihood that the true effect size is lower than the value Chen *et al.* obtained has a further implication, namely that the studies in this literature are almost certainly underpowered. As table 1 shows, the sample sizes in these studies are very small (mean and median per cell = 30). If we take the larger of the two corrected estimates of the mean effect size (i.e. $d = 0.30$ from 3PSM), we can calculate that experiments with these sample sizes have power of only 0.31 to detect (one-tailed) a population effect size of this magnitude. If we take an even more optimistic view and assume that the true effect size is $d = 0.50$, power is still only 0.61.

We do not for one moment underestimate the difficulty of conducting field studies and fully recognize the challenges of collecting data in workplace settings. Nevertheless, before collecting any data, researchers in this area could calculate that with the average sample size they are able to recruit ($N = 30$), achieving a conventional level of power (0.80) would require goal priming to be of a magnitude ($d = 0.65$) that is over twice as large as the effects of ibuprofen on pain relief ($d = 0.28$) and also appreciably larger than the tendency of men to weigh more than women ($d = 0.54$) (estimates from [68]). This level of effect size is implausible, especially bearing in mind that even under the carefully controlled confines of the laboratory, the magnitude of goal priming is barely half this level ($d = 0.31$, from Weingarten *et al.*'s meta-analysis). In the event, it would not be surprising if the effect sizes of some published studies are inflated by sampling error, or that some studies failing to find statistically significant effects go unpublished.

# 7. Conclusion

We share some important points of agreement with Chen *et al.* [23], not least that an overall meta-analysis of goal priming studies on performance measures is potentially illuminating, and that a meta-analysis of the nested set of field studies is of even greater potential significance. Field studies revealing translation of laboratory research into applied settings are challenging to undertake, and hence it is important to use tools such as meta-analysis to ensure that the maximum information is extracted from what is inevitably a fairly small body of studies. When it comes to the implementation of these aims, however, we begin to diverge sharply from Chen *et al.* Our detailed examination of the studies they included in both their larger overall meta-analysis and the meta-analysis restricted to field studies reveals issues concerning some of their inclusion/exclusion decisions. The net effect of these is to undermine the meaningfulness of the meta-analytic effect size estimates they reported.[8]

To remedy this shortcoming in the analysis of field experiments, we collated a modified set of effect sizes that conforms, we argue, to a much more comprehensive and rigorous selection protocol. At first glance supporters of goal priming might find the results of this revised meta-analysis particularly congenial as it reveals an effect size almost as large as that reported by Chen *et al.* However, the clear

---

[6]The exact set of plausible conditions in which each method achieved its worst-case false-positive rate differed across the methods. Thus for *p*-curve the most challenging conjunction is no RDFs combined with medium publication bias, whereas for the random-effects method, it is high RDFs combined with medium publication bias.

[7]In a response to other commentaries on their article, Chen *et al.* [23] acknowledged the importance of testing for publication bias in their overall meta-analysis and reported that a trim-and-fill analysis, while confirming the existence of missing studies, nonetheless yielded a corrected effect size greater than zero. Just as with the field study meta-analysis described in this section, however, trim-and-fill is an invalid correction method for the overall dataset. Carter *et al.*'s [62] simulations show that under conditions that are appropriate for the overall meta-analysis (severity of publication bias = medium, τ = 0.4, RDFs = medium, number of studies = 30), trim-and-fill has a false-positive rate of over 60%. Even if the true effect size was zero, trim-and-fill would frequently conclude otherwise.

[8]Putting aside these issues, it is worth noting that when applied to Chen *et al.*'s set of $k = 8$ field studies, both PET, $M = 0.12$ [−0.90, 1.13] and 3PSM, $M = 0.46$ [0.29, 0.62], generate appreciably lower bias-corrected estimates.

signals of publication bias and/or exploitation of RDF, documented by two bias-correction methods known to provide valid estimates in the range conditions that plausibly apply in this dataset [62], point to a very different conclusion: these studies are biased, either because only significant effects have reached the published literature or because their findings have been inflated by exploitation of RDF. Chen *et al.* estimated the population effect size for goal priming in the field at Cohen's $d = 0.68$ [0.55, 0.81]. We submit that a more credible estimate, based on the PET (PET) and the 3PSM selection model [66], is much smaller than this (if it is greater than zero at all) and that the average power of the experiments in this domain is accordingly likely to be very low. Our findings are aligned with those of Weingarten *et al.* [30] who also found clear evidence of publication bias, including funnel plot asymmetry, in their meta-analysis of experiments that employed verbal goal primes.

We do not wish to overstate our findings. The number of studies in the meta-analysis is modest, and publication bias in this field is an inferred rather than a concretely proven fact. We believe it would be unwise to attach substantial weight to the exact bias-corrected effect size estimates, though of course we suggest that they are more credible because they take rightful account of a property of the studies—publication bias—that Chen's *et al.*'s estimate does not. We are not claiming that goal priming in the workplace does not exist. Instead our more modest conclusion is that the available evidence falls short of demonstrating it to an appropriate level of confidence, and that further research is urgently needed. What requirements should this research fulfil in order to move beyond our current state of knowledge? The answer to this is straightforward: future studies need to be high-powered and pre-registered. Published studies have average sample sizes that are so small that even if goal priming is a genuine medium-sized effect (approximately equal to 0.5), their success in rejecting the null hypothesis would be little better than a coin flip. Moreover, studies with such small samples are unable to estimate the effect size with any precision. This is starkly illustrated in figure 1 where, for example, it can be seen that the 95% CI in one study's estimate of the priming effect covered the range from 0.06 to 1.84 ([53], Study 1). In addition to much higher power, future studies should be pre-registered to protect against any suspicion of *p*-hacking and published regardless of the results to protect against publication bias [18].

Data accessibility. Experimental data and analysis code are publicly available via the Open Science Framework (OSF) at https://osf.io/5cjzp/.

Authors' contribution. D.S. and M.V. carried out the literature search, data analysis and drafted the manuscript; all authors gave final approval for publication and agree to be held accountable for the work performed therein.

Competing interests. We declare we have no competing interests.

Funding. This research was supported by grants PSI2017–85159-P (AEI/FEDER, UE), 2016-T1/SOC-1395 (Comunidad de Madrid, Programa de Atracción de Talento Investigador), and ES/P009522/1 (United Kingdom Economic and Social Research Council).

Acknowledgements. We thank Evan Carter for his helpful suggestions.

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
