## [Peer Review File · Royal Society Open Science]

Review History

RSOS-210544.R0 (Original submission)

Review form: Reviewer 1 (Lincoln Colling)

Is the manuscript scientifically sound in its present form?

Yes

Are the interpretations and conclusions justified by the results?

Yes

Is the language acceptable?

Yes

Do you have any ethical concerns with this paper?

No

Have you any concerns about statistical analyses in this paper?

No

Recommendation?

Accept with minor revision (please list in comments)

Comments to the Author(s)

I really enjoyed the manuscript and thought it provide a well balanced critique of the commented on work. I'm personally rather skeptical of bias correction methods and so I appreciate that the authors did not overstate their findings. By highlighting both methodological (PRISMA, inclusion criteria) and statistical shortcomings, I thought the authors did a good job in presenting a more full case than Chen et al, and therefore I think it makes a valuable contribution to the literature.

For this reason, I will restrict my comments to some omissions on the part of the authors. The primary one is that the dataset that has been made available on the OSF is incomplete. Specifically, the dataset is missing the columns than are needed to more the moderator analysis and therefore I was unable to reproduce any of these analyses. These columns should be included.

Furthermore, I would also argue the authors to make use of, for example, the checkpoint package in their supplied R script. I've included an example of what this might look like below. This is needed to ensure reproducibility of their results, because it would allow them to fix the package versions and R versions to the ones that were actually used to perform the analysis (as R package versions change their outputs can also sometimes change). Related to this, I'd also urge the authors to include the file `2-p-curve.R` on their OSF page. The CC-BY-4 licence that this file is published under would allow them to do this.

Finally, turning back to the manuscript itself, could the authors double check their total N, because I get a different number.

```

```r
see https://cran.r-project.org/web/packages/checkpoint/vignettes/checkpoint.html
library(checkpoint)
checkpoint("2020-01-01", r_version="4.0.0") # replace with desired date and R version
```

```

Review form: Reviewer 2 (Thom Baguley)

Is the manuscript scientifically sound in its present form?

Yes

Are the interpretations and conclusions justified by the results?

Yes

Is the language acceptable?

Yes

Do you have any ethical concerns with this paper?

No

Have you any concerns about statistical analyses in this paper?

Yes

Recommendation?

Accept with minor revision (please list in comments)

Comments to the Author(s)

Overall, this is a really well written paper with careful arguments, analysis and appropriate conclusions. Most of my comments are very minor, but there is one area where I think the analyses fall short and could be improved. I don't think this will change the outcome (actually it might further clarify what is going on). I very like the general approach of using bias correction approaches as a sensitivity check and some other nice touches (including checking for non-independent studies).

The main potential concern is whether the authors mix within and between effect sizes. I'm sure meta-analyses do this all the time but it isn't clear to me how they've addressed this. (Assuming I'm right; the text descriptions of the effects in the original studies are ambiguous and a quick search of the original papers didn't clarify as I ran into a paywall accessing one the original articles. One issue is the authors may have referred to a mixed ANOVA as a repeated measures ANOVA). The potential issue is that standardised effect sizes from from within designs are not directly comparable to those from between designs without adjustment.

Morris, S. B., & DeShon, R. P. (2002). Combining effect size estimates in meta-analysis with repeated measures and independent-groups designs. *Psychological methods*, 7(1), 105-125.

As part of this I'd include between/within design as a moderator as it looks like the very largest effects are from designs that might be within-participants.

A second issue is that I couldn't reproduce all the analyses. Specifically the moderator analyses. It seems that the data file at OSF doesn't code up the moderators. I also didn't have time to check the regression tests with external functions, but the effects look plausible (but it would have been useful to include the functions in the same file or to use `source()` to pull them in from an external file).

Further minor comments (page numbers refer to the pdf not the ms numbers):

p.4 lines 7-9 The generic reference to "Priming research ..." isn't helpful. Certain types of priming are very robust. Terminology here is a bit fraught but some distinction to clarify the types of priming effects that are of concern such as "behavioral" priming would be helpful.

p.5 lines 14-16 Is it worth stating that this is the pattern expected if small NS effects are harder to publish? (Obvious to many but not all readers)

p.5 line 31 "priming" used generically again - could clarify here or state earlier that that's what you mean throughout?

p.7 lines 7-9 The default assumption should be that publication bias is present in my view in any literature where publication is filtered by a decision threshold such as a p value. The exceptions need to be supported by plausible information (e.g., a meta-analysis of a trials register or other audited complete set of studies or a meta-analysis of a secondary/incidental finding that wouldn't be filtered by publication bias). The issue is really the degree of bias - and that's why sensitive checks are potentially useful.

p.7 lines 7-12 - you are very careful in the conclusions to caution against using a corrected estimate naively. I therefore don't like this phrasing. Generally we don't know enough to accurately correct the effect size estimate when bias is present. I think the best we can do in most cases is to use them as sensitivity checks and for benchmarking ranges of effect that can't be ruled out. I think this is precisely what you do - so it would be helpful to rephrase this. especially this part: "the meta-analytic effect size needs to be adjusted".

p.7 lines 24-26. Very pedantic, but I'm not sure about this: "did not explain the method they adopted for combining results (fixed- effect vs. random effects)". I don't think random or fixed effects are the methods for combining results. They are the type of statistical model. For instance you can combine weighted or unweighted with fixed effects (or potentially other wise).

p.13 lines 39-44. I think this is fair, but $k = 13$ is limiting in addressing moderator effects and precision of estimating heterogeneity and between study variance. It also limits power to detect asymmetry or bias (but that's not such an issue here, though it is for the Chen study).

p.13 lines 24-25 " $d = 0.64$, 95% CI [0.41, 0.88]" You get slightly wider CIs with the Knapp-Hartung adjustment using `test="knha"` in the `rma()` function. Largely immaterial here but further lowers the precision relative to the Chen et al. MA.

p.15-16 discussion of causes of asymmetry omits the heterogeneity, which can produce asymmetry quite easily.

p.17 lines 3-4 "If the 'missing' effects were added to the figure, the association would disappear." I don't like this phrasing as it suggests that this practice is a good idea (which I don't think it what you intend). "The likely missing studies would almost certainly change the pattern of effects if they could be found" or similar might be better.

p.19 lines 5-16 - I'd reorder this section to make the tradeoff more explicit as 20% false positives is very high. I think satisfactory is also a poor label "satisfactory in this context" might be better. Basically the argument is that false negative rates are very high so false positive rates can't be easily controlled.

p.19 line 60 - I'd say it fits slightly better.

p.20 lines 21-24 Can you rephrase make to make clear that the 63% false positive rate is known known with certainty. "Given that a false positive of 60% or so is likely ..." or similar

p.21 lines 13-20 "acceptable level of power (0.80)" This is a common convention, but its not reasonable to say that's an acceptable level of power without qualification. I'd argue that with $\alpha = .05$ 95% power is acceptable and 80% merely a convenient practical target. (This is a digression but my point is that we need to stop signalling arbitrary thresholds like this as good practice).

Decision letter (RSOS-210544.R0)

Dear Professor Shanks

On behalf of the Editors, we are pleased to inform you that your Manuscript RSOS-210544 "Publication Bias and Low Power in Field Studies on Goal Priming" has been accepted for publication in Royal Society Open Science subject to minor revision in accordance with the referees' reports. Please find the referees' comments along with any feedback from the Editors below my signature.

Please submit your revised manuscript and required files (see below) no later than 7 days from today's (ie 02-Sep-2021) date. Note: the ScholarOne system will 'lock' if submission of the revision is attempted 7 or more days after the deadline. If you do not think you will be able to meet this deadline please contact the editorial office immediately.

on behalf of Professor Zoltan Dienes (Associate Editor) and Essi Viding (Subject Editor)
openscience@royalsociety.org

Associate Editor Comments to Author (Professor Zoltan Dienes):

Associate Editor: 1

Comments to the Author:

Sorry for the delay in getting back to you. It was somewhat delayed in getting to me; and then I asked over twenty potential reviewers - but it was worth it because the two who got back provided thoughtful and careful reviews. Both are very positive about the manuscript, but with important points that need addressing. Both mention the lack of coding of moderator variables in the provided data. Each makes other insightful points as well. I look forward to your response dealing with these issues.

Reviewer comments to Author:

Reviewer: 1

Comments to the Author(s)

I really enjoyed the manuscript and thought it provide a well balanced critique of the commented on work. I'm personally rather skeptical of bias correction methods and so I appreciate that the authors did not overstate their findings. By highlighting both methodological (PRISMA, inclusion criteria) and statistical shortcomings, I thought the authors did a good job in presenting a more full case than Chen et al, and therefore I think it makes a valuable contribution to the literature.

For this reason, I will restrict my comments to some omissions on the part of the authors. The primary one is that the dataset that has been made available on the OSF is incomplete.

Specifically, the dataset is missing the columns than are needed to more the moderator analysis and therefore I was unable to reproduce any of these analyses. These columns should be included.

Furthermore, I would also argue the authors to make use of, for example, the checkpoint package in their supplied R script. I've included an example of what this might look like below. This is needed to ensure reproducibility of their results, because it would allow them to fix the package versions and R versions to the ones that were actually used to perform the analysis (as R package versions change their outputs can also sometimes change). Related to this, I'd also urge the authors to include the file `2-p-curve.R` on their OSF page. The CC-BY-4 licence that this file is published under would allow them to do this.

Finally, turning back to the manuscript itself, could the authors double check their total N, because I get a different number.

```
```r
see https://cran.r-project.org/web/packages/checkpoint/vignettes/checkpoint.html
library(checkpoint)
checkpoint("2020-01-01", r_version="4.0.0") # replace with desired date and R version
```
```

Reviewer: 2

Comments to the Author(s)

Overall, this is a really well written paper with careful arguments, analysis and appropriate conclusions. Most of my comments are very minor, but there is one area where I think the analyses fall short and could be improved. I don't think this will change the outcome (actually it might further clarify what is going on). I very like the general approach of using bias correction approaches as a sensitivity check and some other nice touches (including checking for non-independent studies).

The main potential concern is whether the authors mix within and between effect sizes. I'm sure meta-analyses do this all the time but it isn't clear to me how they've addressed this. (Assuming I'm right; the text descriptions of the effects in the original studies are ambiguous and a quick search of the original papers didn't clarify as I ran into a paywall accessing one the original articles. One issue is the authors may have referred to a mixed ANOVA as a repeated measures ANOVA). The potential issue is that standardised effect sizes from from within designs are not directly comparable to those from between designs without adjustment.

Morris, S. B., & DeShon, R. P. (2002). Combining effect size estimates in meta-analysis with repeated measures and independent-groups designs. *Psychological methods*, 7(1), 105-125.

As part of this I'd include between/within design as a moderator as it looks like the very largest effects are from designs that might be within-participants.

A second issue is that I couldn't reproduce all the analyses. Specifically the moderator analyses. It seems that the data file at OSF doesn't code up the moderators. I also didn't have time to check the regression tests with external functions, but the effects look plausible (but it would have been useful to include the functions in the same file or to use source() to pull them in from an external file).

Further minor comments (page numbers refer to the pdf not the ms numbers):

p.4 lines 7-9 The generic reference to "Priming research ..." isn't helpful. Certain types of priming are very robust. Terminology here is a bit fraught but some distinction to clarify the types of priming effects that are of concern such as "behavioral" priming would be helpful.

p.5 lines 14-16 Is it worth stating that this is the pattern expected if small NS effects are harder to publish? (Obvious to many but not all readers)

p.5 line 31 "priming" used generically again - could clarify here or state earlier that that's what you mean throughout?

p.7 lines 7-9 The default assumption should be that publication bias is present in my view in any literature where publication is filtered by a decision threshold such as a p value. The exceptions need to be supported by plausible information (e.g., a meta-analysis of a trials register or other audited complete set of studies or a meta-analysis of a secondary/incidental finding that wouldn't be filtered by publication bias). The issue is really the degree of bias - and that's why sensitive checks are potentially useful.

p.7 lines 7-12 - you are very careful in the conclusions to caution against using a corrected estimate naively. I therefore don't like this phrasing. Generally we don't know enough to accurately correct the effect size estimate when bias is present. I think the best we can do in most cases is to use them as sensitivity checks and for benchmarking ranges of effect that can't be ruled out. I think this is precisely what you do - so it would be helpful to rephrase this. especially this part: "the meta-analytic effect size needs to be adjusted".

p.7 lines 24-26. Very pedantic, but I'm not sure about this: "did not explain the method they adopted for combining results (fixed- effect vs. random effects)". I don't think random or fixed effects are the methods for combining results. They are the type of statistical model. For instance you can combine weighted or unweighted with fixed effects (or potentially other wise).

p.13 lines 39-44. I think this is fair, but $k = 13$ is limiting in addressing moderator effects and precision of estimating heterogeneity and between study variance. It also limits power to detect asymmetry or bias (but that's not such an issue here, though it is for the Chen study).

p.13 lines 24-25 "d = 0.64, 95% CI [0.41, 0.88]" You get slightly wider CIs with the Knapp-Hartung adjustment using `test="knha"` in the `rma()` function. Largely immaterial here but further lowers the precision relative to the Chen et al. MA.

p.15-16 discussion of causes of asymmetry omits the heterogeneity, which can produce asymmetry quite easily.

p.17 lines 3-4 "If the 'missing' effects were added to the figure, the association would disappear." I don't like this phrasing as it suggests that this practice is a good idea (which I don't think it what you intend). "The likely missing studies would almost certainly change the pattern of effects if they could be found" or similar might be better.

p.19 lines 5-16 - I'd reorder this section to make the tradeoff more explicit as 20% false positives is very high. I think satisfactory is also a poor label "satisfactory in this context" might be better. Basically the argument is that false negative rates are very high so false positive rates can't be easily controlled.

p.19 line 60 - I'd say it fits slightly better.

p.20 lines 21-24 Can you rephrase make to make clear that the 63% false positive rate is known known with certainty. "Given that a false positive of 60% or so is likely ..." or similar

p.21 lines 13-20 "acceptable level of power (0.80)" This is a common convention, but its not reasonable to say that's an acceptable level of power without qualification. I'd argue that with $\alpha = .05$ 95% power is acceptable and 80% merely a convenient practical target. (This is a digression but my point is that we need to stop signalling arbitrary thresholds like this as good practice).

===PREPARING YOUR MANUSCRIPT===

- one version identifying all the changes that have been made (for instance, in coloured highlight, in bold text, or tracked changes);
- a 'clean' version of the new manuscript that incorporates the changes made, but does not highlight them. This version will be used for typesetting.

===PREPARING YOUR REVISION IN SCHOLARONE===

Please ensure that you include a summary of your paper at Step 2 'Type, Title, & Abstract'. This should be no more than 100 words to explain to a non-scientific audience the key findings of

your research. This will be included in a weekly highlights email circulated by the Royal Society press office to national UK, international, and scientific news outlets to promote your work.

Author's Response to Decision Letter for (RSOS-210544.R0)

See Appendix A.

Decision letter (RSOS-210544.R1)

Dear Professor Shanks,

It is a pleasure to accept your manuscript entitled "Publication Bias and Low Power in Field Studies on Goal Priming" in its current form for publication in Royal Society Open Science. The comments from the Editors are included at the foot of this letter.

on behalf of Professor Zoltan Dienes (Associate Editor) and Essi Viding (Subject Editor)
openscience@royalsociety.org

Associate Editor Comments to Author (Professor Zoltan Dienes):

You have addressed the comments of the reviewers well, and I can now accept the paper. I hope the paper has due impact on the goal priming literature.

Appendix A

| | |
|--|--|
| Reviewer 1 | |
| I really enjoyed the manuscript and thought it provide a well balanced critique of the commented on work. I'm personally rather skeptical of bias correction methods and so I appreciate that the authors did not overstate their findings. By highlighting both methodological (PRISMA, inclusion criteria) and statistical shortcomings, I thought the authors did a good job in presenting a more full case than Chen et al, and therefore I think it makes a valuable contribution to the literature. | We appreciate these positive comments. |
| For this reason, I will restrict my comments to some omissions on the part of the authors. The primary one is that the dataset that has been made available on the OSF is incomplete. Specifically, the dataset is missing the columns than are needed to more the moderator analysis and therefore I was unable to reproduce any of these analyses. These columns should be included. | We apologise that the uploaded dataset was an earlier version that lacked the moderator coding. This has now been corrected and a new version uploaded to OSF. |
| Furthermore, I would also argue the authors to make use of, for example, the checkpoint package in their supplied R script. I've included an example of what this might look like below. This is needed to ensure reproducibility of their results, because it would allow them to fix the package versions and R versions to the ones that were actually used to perform the analysis (as R package versions change their outputs can also sometimes change). Related to this, I'd also urge the authors to include the file `2-p-curve.R` on their OSF page. The CC-BY-4 licence that this file is published under would allow them to do this. Finally, turning back to the manuscript itself, could the authors double check their total N, because I get a different number. <pre> ```r # see https://cran.r-project.org/web/packages/checkpoint/vignettes/checkpoint.html library(checkpoint) checkpoint("2020-01-01", r_version="4.0.0") # replace with desired date and R version ``` </pre> | We are grateful for this suggestion and have added relevant checkpoint code to the R script. The updated version of the script downloads the “2-p-curve.R” file directly from the original OSF project by Carter et al. This circumvents the problem of uploading to our project a piece of code that was authored by other researchers. The reviewer has presumably added up all the N's for the relevant groups in the data file ($N = 790$). The reason this is different from the figure we cite ($N = 683$) is because in 3 studies, a single control group was compared against two independent experimental groups. We have clarified this in footnote 3. |
| Reviewer 2 | |
| Overall, this is a really well written paper with careful arguments, analysis and appropriate conclusions. Most of my comments are very minor, but there is one area where I think the analyses fall short and could be improved. I don't think this will change the outcome (actually it might further clarify what is going on). I very like the general approach of using bias correction approaches as a sensitivity check and some other nice touches (including checking for non-independent studies). | We appreciate these positive comments. |

| | |
|---|---|
| The main potential concern is whether the authors mix within and between effect sizes. I'm sure meta-analyses do this all the time but it isn't clear to me how they've addressed this. (Assuming I'm right; the text descriptions of the effects in the original studies are ambiguous and a quick search of the original papers didn't clarify as I ran into a paywall accessing one the original articles. One issue is the authors may have referred to a mixed ANOVA as a repeated measures ANOVA). The potential issue is that standardised effect sizes from from within designs are not directly comparable to those from between designs without adjustment. Morris, S. B., & DeShon, R. P. (2002). Combining effect size estimates in meta-analysis with repeated measures and independent-groups designs. Psychological methods, 7(1), 105-125. As part of this I'd include between/within design as a moderator as it looks like the very largest effects are from designs that might be within-participants. | This issue is easily dealt with – all experiments used a between-subjects design. We have added an explicit statement to this effect on p12. |
| A second issue is that I couldn't reproduce all the analyses. Specifically the moderator analyses. It seems that the data file at OSF doesn't code up the moderators. I also didn't have time to check the regression tests with external functions, but the effects look plausible (but it would have been useful to include the functions in the same file or to use source() to pull them in from an external file). | See comment above on this issue. |
| p.4 lines 7-9 The generic reference to "Priming research ..." isn't helpful. Certain types of priming are very robust. Terminology here is a bit fraught but some distinction to clarify the types of priming effects that are of concern such as "behavioral" priming would be helpful. | We have modified 2 uses of the term 'priming' in the opening paragraph (p3) to 'behaviour priming'. |
| p.5 lines 14-16 Is it worth stating that this is the pattern expected if small NS effects are harder to publish? (Obvious to many but not all readers) | We have made this point clearer in a new sentence added on p4. |
| p.5 line 31 "priming" used generically again - could clarify here or state earlier that that's what you mean throughout? | We believe that the clarification on p3 (see above) deals with this concern. |
| p.7 lines 7-9 The default assumption should be that publication bias is present in my view in any literature where publication is filtered by a decision threshold such as a p value. The exceptions need to be supported by plausible information (e.g., a meta-analysis of a trials register or other audited complete set of studies or a meta-analysis of a secondary/incidental finding that wouldn't be filtered by publication bias). The issue is really the degree of bias - and that's why sensitive checks are potentially useful. | No response required. |
| p.7 lines 7-12 - you are very careful in the conclusions to caution against using a corrected estimate naively. I therefore don't like this phrasing. Generally we don't know enough to accurately correct the effect size estimate when bias is present. I think the | We appreciate the suggestion to use the term 'sensitivity' check and have added this on p6. However we also prefer to |

| | |
|--|---|
| best we can do in most cases is to use them as sensitivity checks and for benchmarking ranges of effect that can't be ruled out. I think this is precisely what you do - so it would be helpful to rephrase this. especially this part: "the meta-analytic effect size needs to be adjusted". | retain the phrasing regarding 'correcting an effect-size estimate', which is standard terminology. |
| p.7 lines 24-26. Very pedantic, but I'm not sure about this: "did not explain the method they adopted for combining results (fixed- effect vs. random effects)". I don't think random or fixed effects are the methods for combining results. They are the type of statistical model. For instance you can combine weighted or unweighted with fixed effects (or potentially other wise). | On p6 we have replaced 'explain the method' with 'describe the statistical model'. |
| p.13 lines 39-44. I think this is fair, but $k = 13$ is limiting in addressing moderator effects and precision of estimating heterogeneity and between study variance. It also limits power to detect asymmetry or bias (but that's not such an issue here, though it is for the Chen study). | We have added a comment on p12 about the sample size limiting the ability to explore moderators. |
| p.13 lines 24-25 "$d = 0.64$, 95% CI [0.41, 0.88]" You get slightly wider CIs with the Knapp-Hartung adjustment using <code>test="knha"</code> in the <code>rma()</code> function. Largely immaterial here but further lowers the precision relative to the Chen et al. MA. | Because this adjustment is not well-known we prefer to retain the estimate given in the text. |
| p.15-16 discussion of causes of asymmetry omits the heterogeneity, which can produce asymmetry quite easily. | This is now acknowledged on p14. |
| p.17 lines 3-4 "If the 'missing' effects were added to the figure, the association would disappear." I don't like this phrasing as it suggests that this practice is a good idea (which I don't think it what you intend). "The likely missing studies would almost certainly change the pattern of effects if they could be found" or similar might be better | The suggested rephrasing has been adopted on p16. |
| p.19 lines 5-16 - I'd reorder this section to make the tradeoff more explicit as 20% false positives is very high. I think satisfactory is also a poor label "satisfactory in this context" might be better. Basically the argument is that false negative rates are very high so false positive rates can't be easily controlled. | We have adopted the suggested re-ordering (p.18) and added 'in this context'. |
| p.19 line 60 - I'd say it fits slightly better. | The word 'slightly' is quite evaluative. Given that the effect is significant at $p < .02$, we prefer to retain the existing phrasing. |
| p.20 lines 21-24 Can you rephrase make to make clear that the 63% false positive rate is known known with certainty. "Given that a false positive of 60% or so is likely ..." or similar | The suggested rephrasing has been adopted on p19. |
| p.21 lines 13-20 "acceptable level of power (0.80)" This is a common convention, but its not reasonable to say that's an acceptable level of power without qualification. I'd argue that with $\alpha = .05$ 95% power is acceptable and 80% merely a convenient practical target. (This is a digression but my point is that we need to stop signalling arbitrary thresholds like this as good practice). | We have replaced 'acceptable' with 'conventional' |